# Germinated Buckwheat: Effects of Dehulling on Phenolics Profile and Antioxidant Activity of Buckwheat Seeds

**DOI:** 10.3390/foods10040740

**Published:** 2021-04-01

**Authors:** Andrej Živković, Tomaž Polak, Blaž Cigić, Tomaž Požrl

**Affiliations:** Department of Food Science and Technology, Biotechnical Faculty, University of Ljubljana, SI-1111 Ljubljana, Slovenia; andrej.zivkovic@bf.uni-lj.si (A.Ž.); tomaz.polak@bf.uni-lj.si (T.P.); blaz.cigic@bf.uni-lj.si (B.C.)

**Keywords:** buckwheat, dehulling, germination, LC-MS, free phenolic, bound phenolic, antioxidant activity

## Abstract

The aim was to investigate the effects of the cold dehulling of buckwheat seeds on their germination, total phenolic content (TPC), antioxidant activity (AA) and phenolics composition. Cold dehulling had no negative effects on germination rate and resulted in faster rootlet growth compared to hulled seeds. Although the dehulling of the seeds significantly decreased TPC and AA, the germination of dehulled seeds resulted in 1.8-fold and 1.9-fold higher TPC and AA compared to hulled seeds. Liquid chromatography coupled to mass spectrometry identified several phenolic compounds in free and bound forms. Rutin was the major compound in hulled seeds (98 µg/g dry weight), orientin and vitexin in 96-h germinated dehulled seeds (2205, 1869 µg/g dry weight, respectively). During germination, the increases in the major phenolic compounds were around two orders of magnitude, which were greater than the increases for TPC and AA. As well as orientin and vitexin, high levels of other phenolic compounds were detected for dehulled germinated seeds (e.g., isoorientin, rutin; 1402, 967 µg/g dry weight, respectively). These data show that dehulled germinated seeds of buckwheat have great potential for use in functional foods as a dietary source of phenolic compounds with health benefits.

## 1. Introduction

The germination of edible seeds is a recognized method for improving the nutritional value of seeds [1,2,3,4]. Malted grains have traditionally been used in beer production and in recent years, as a functional ingredient in the baking industry, as groats and flour. The use of the germinated seeds of many plant species is increasing due to the often high contents and availability of nutrients compared to dry seeds, and to the associated beneficial effects on human health. In recent years, there have been many studies on the effects of the germination of cereals and pseudocereals, and their various health benefits for the prevention of chronic diseases, such as heart disease, cancers and diabetes [5,6,7,8,9]. Dynamic changes during germination can lead to the breakdown of macronutrients, such as carbohydrates, proteins and lipids. In addition, the levels of polyphenols, vitamins and other bioactive compounds rapidly increase during germination due to de novo synthesis and transformation, thus enhancing the health-promoting effects of seeds [10,11,12,13].

Buckwheat (*Fagopyrum esculentum*) is a pseudocereal that is highly adaptable to harsh environmental conditions, has a short growth period, and shows greater resistance to pests compared to other cereals. Nowadays, buckwheat is recognized as an important gluten-free functional food ingredient due to its balanced composition of macronutrients and high content of bioactive compounds. The protein content in buckwheat seeds is similar to that of wheat grain, but with a higher biological value due to the balanced, lysine-rich amino-acid composition and the low content of storage prolamins [14]. Buckwheat seeds have a high levels of phenolic compounds, and especially flavonoids, such as rutin [15], orientin, vitexin and isovitexin, which have been reported to have several health benefits [16,17,18,19]. Different studies in humans have shown associations between the consumption of buckwheat seed products with antihypertensive effects [20] and cholesterol-lowering effects [21]. However, the total phenolic content (TPC) in cereals has often been underestimated, as significant proportions of these phenolic compounds are covalently bound to cell wall materials, and thus need to be extracted by alkaline hydrolysis. For the correct estimation of TPC, this portion of bound phenolics must also be considered [22].

Buckwheat seeds are usually consumed in the form of products made from their flour, or as the dehulled seeds, known as groats. Almost all of the data available on the TPC of germinated buckwheat seeds refer to hulled seeds, and although the hull is not edible, it is an important contributor to TPC [23,24]. The usual processing method used for groats production is thermal dehulling, where the seeds are first boiled or steamed at high temperatures of 130 °C to 160 °C, under high pressure. They are then cooled, conditioned and dried prior to dehulling. During this process, the seed endosperm swells and the husk breaks so that it can be easily removed [25,26]. The thermal processing used in the production of groats significantly reduces the nutritional value of the seeds, and from the point of view of sprout production, this reduces the seed germination. Several studies have shown the effects of thermal dehulling, in terms of decreases in TPC, antioxidant activity (AA) and the major flavonoids, rutin and quercetin [27,28].

Buckwheat has been used previously for the preparation of sprouts and microgreens [29], which are generally considered to be a rich source of bioactive compounds. For the production of buckwheat sprouts ready for consumption, it is nevertheless essential to obtain the groats, with the need to thus remove the seed hull. To retain high germination rates, the typical thermal processing method for the seeds is not an option. In comparison to other hulled cereals, buckwheat seeds are easily damaged during dehulling, and hence the obtainment of intact groats with high germination rates is a challenging task.

However, the many reported advantages of hull-less buckwheat seeds justify the efforts needed to obtain this product for use as a functional food ingredient. A previous study on oat [30] showed that germination of the dehulled grain resulted in higher TPC and AA in comparison to the grain with the hulls. It can be assumed that similar effects will apply to buckwheat seeds, which would justify the efforts to prepare hull-less seeds that retain high germination rates and can be used as a functional food ingredient.

Germination is initiated by increasing the moisture content of grain or seeds to 43% to 45% by soaking in water [31]. Upon the initiation of germination, enzyme synthesis and kernel modification take place, and the seeds rapidly undergo metabolic activities that include respiration, nutrient degradation and secondary metabolite synthesis [32,33]. Several studies have investigated the effects of germination on the TPC, AA and phenolics composition of buckwheat seeds [34,35]. 

In our study, a new processing approach of mechanical—the cold dehulling method—was used for the buckwheat, which maintained a high germination rate. Previous studies and other scientific articles have focused on the effects of germination on phenolic content and antioxidant activity of hulled germinated buckwheat. However, to the best of our knowledge, there have not been any studies into the effects of buckwheat seed dehulling on the germination bioactivities. Furthermore, for the Slovenian domestical cultivar of common buckwheat (*Fagopyrum esculentum*) “Čebelica” [36] used here, there appears to be no information on their phenolic profile.

The objectives of this study were: (1) to determine the effects of the dehulling of buckwheat seeds on their growth (dry weight, rootlet length); (2) to study the dynamic changes in the TPC and AA of extracts of hulled and dehulled seeds during germination; and (3) to characterise the dynamic changes in the content of individual phenolic compounds in their free and bound forms at different stages of germination.

## 2. Materials and Methods

### 2.1. Materials

The buckwheat seeds were purchased from a local producer (Krasinec, Slovenia) and grown under organic growing conditions. Methanol, sodium hydroxide, hydrochloric acid, sodium bicarbonate, 2,2′-azino-bis (3-ethylbenzothiazolin-6-sulfonic acid diammonium salt) (ABTS reagent), 1,1-diphenyl-2-picrylhydrazyl (DPPH), and gallic acid were from Sigma-Aldrich (Steinheim, Germany). Folin–Ciocalteu reagent was from Merck (Darmstadt, Germany), and manganese dioxide was from Kemika (Zagreb, Croatia). The analytical standards of rutin (PN:78095-25MG-F), orientin (PN:09765-1MG), isoorientin (PN:78109-5MG), vitexin (PN:49513-10MG-F), catechin (PN:43412-10MG), epicatechin (PN:68097-10MG) and p-coumaric acid (PN:C9008-10G) were from Sigma–Aldrich (Steinheim, Germany), and hyperoside (PN:0018-05-85) was from HWI Analytik (Rülzheim, Germany). Along with the compounds identified, eight other phenolic standards were used: caffeic acid (PN:C0625-5G), gallic acid (PN:91215-100MG), luteolin-7-glucoside (PN:49968-10MG), naringenin (PN:N5893-5G), morin (PN:M4008-5G), quercetin (PN:Q4951-10G) and kaempferol (PN:K0133-10MG), which were all from Sigma-Aldrich (Steinheim, Germany), and chlorogenic acid (PN:0050-05-90), which was from HWI Analytik (Rülzheim, Germany). All of the standards used were analytical or HPLC grade. All aqueous solutions were prepared using Milli-Q purified water (Merck Millipore, Bedford, MA, USA).

### 2.2. Cold Dehulling of Buckwheat Seeds

A “cold dehulling” method was used to dehull the seeds in this study, whereby the groats obtained appeared to show good germination rates. Prior to dehulling, the hulled seeds were fractionated using a series of square-holed sieves with mesh sizes defined as 5.5, 6.0 and 6.5 (3.70, 3.35, 3.00 mm). Approximately 15% of the seeds (diameter >3.7 mm, <2.7 mm) were discarded, and so not used further in this study. The remaining seed fractions of different sizes were dehulled separately using a stone mill (TYP A130; Osttiroler Grain Mills, Austria) [37] with a gap between the stones of 3.0 mm, 3.4 mm and 3.8 mm, respectively. The hulls were separated from the groats using a flow of air. The dehulled seeds were separated using the respective sieves, and for the remaining hulled seeds, the dehulling was repeated.

### 2.3. Germination 

Before germination, the hulled and dehulled seeds were soaked in water for 8 h, with a 15-min period out of the water every hour. After soaking, all of the seeds were placed in a thin layer on moistened filter paper in glass Petri dishes. Germination was carried out in a 20 °C thermostated growth chamber, with moistened filter papers to ensure high humidity (relative humidity, >95%). The filter papers were kept moist by spraying with distilled water as needed. The sprouts were harvested for analysis at 24, 48, 72 and 96 h after the start of soaking.

### 2.4. Determination of Total Dry Mass Loss during Germination

Previous studies have shown that during germination, the energy reserves in the form of starch, lipids, and to a certain extent, proteins are mobilised, and become depleted [38,39]. As all of the data here are presented on a dry weight basis, the characterisationof these losses was of crucial importance. For this purpose, a method for estimating the loss of dry matter during germination was developed. Briefly, 100 seeds from each batch with the determined moisture content were placed in glass Petri dishes, and the weight of each batch was determined using laboratory scales. Germination was initiated as described above, and after 24, 48, 72 and 96 h the germinated seeds in the Petri dishes were dried to constant mass in a laboratory dryer. The loss of dry matter was expressed as the difference between the calculated dry matter mass of the initial seeds and the mass determined for the dry germinated seeds.

### 2.5. Extraction of Free Phenolic Compounds

Samples of the seeds and the germinated seeds (i.e., sprouts) were frozen in liquid nitrogen and milled in a laboratory mill (A11 Basic; IKA-Werke, Staufen, Germany). The free phenolic compounds were extracted using 70% aqueous methanol. Briefly, 0.5 g of ground seeds were mixed with 3.0 mL of 70% aqueous methanol, and the mixture was shaken in the dark at room temperature for 40 min, at 200 rpm (EV-403; Tehtnica Železniki, Slovenia). After centrifugation at 8709× *g* for 8 min at 10 °C (Avanti JXN-26; Beckman Coulter, Krefeld, Germany), the supernatant was removed and saved, and the extraction was repeated twice more. The three supernatants were pooled, diluted to 10 mL with 70% aqueous methanol, filtered using 0.45-µm pore size syringe filters (Chromafil A-45/25; cellulose acetate, hydrophilic membrane; Macherey-Nagel, Düren, Germany), and stored at 2 °C until the determination of the TPC, and DPPH and ABTS analysis, within 24 h.

### 2.6. Extraction of Bound Phenolic Compounds

After the methanol extraction, the solid residues were hydrolysed with sodium hydroxide, as described previously [40]. Then, 20 mL of 2 M NaOH was added to the reaction tube, and the mixture was shaken in the dark at room temperature for 4 h, at 200 rpm (Tehtnica Železniki EV-403, Slovenia). The hydrolysed mixture was acidified to pH 3.2 to 3.4 by the addition of 3.5 mL of concentrated formic acid. After centrifugation at 8709× *g* for 8 min at 10 °C (Avanti JXN-26; Beckman Coulter, Krefeld, Germany), the supernatant was removed and filtered through 0.45-µm pore size syringe filters (Chromafil A-45/25; cellulose acetate, hydrophilic membrane; Macherey-Nagel, Düren, Germany) and stored at 2 °C until the determination of the TPC, and DPPH and ABTS analysis, within 24 h.

### 2.7. Total Phenolic Content 

The TPC of the seed extracts was determined using the Folin–Ciocalteu spectrophotometric method described previously [41], with some modifications. Briefly, 0.1 mL of each extract was dispensed into 2.0 mL microcentrifuge tubes and mixed with 1.3 mL of Milli-Q water and 0.3 mL 1:3 diluted of Folin–Ciocalteu phenolic reagent, and allowed to react for 5 min. Then, 0.3 mL of 20% (*w/v*) aqueous Na_2_CO_3_ was added, and after 1 h at room temperature, the absorbances were measured at 765 nm (UV–Vis spectrophotometer; model 8453; Agilent Technologies, Santa Clara, CA, USA). The measurements were compared with a standard curve of a gallic acid (GA) solution, and the TPC is expressed as mg gallic acid equivalents (GAE) per g dry weight of the seed sample (mg GAE/g DW).

### 2.8. DPPH Radical Scavenging Activity 

The DPPH radical scavenging activity of the extracts was determined according to a method described previously [41], with some modifications. Briefly, 50.0 µL of each extract was dispensed into 2.0 mL microcentrifuge tubes, and 250 µL of acetic buffer was added, and the volume was adjusted to 1.0 mL with methanol. Finally, 1 mL of 0.2 mM methanol solution of DPPH was added. The mixture was shaken and then left in the dark for 1 h. The absorbance was then measured at 517 nm (UV–Vis spectrophotometer; model 8453; Agilent Technologies, Santa Clara, CA, USA). A lower absorption of the reaction mixture indicates higher free radical scavenging activity. The absorbance of the control was achieved by replacing the seed sample with methanol. The measurement was compared to a standard curve of a Trolox solution, and the radical scavenging activity is expressed as mg Trolox equivalents per g dry matter (mg TE/g DW).

### 2.9. ABTS Radical Cation Scavenging Activity 

The radical scavenging activities of the seed extracts against the ABTS radical cation were determined according to a method described previously [34], with some modifications. The ABTS stock solution was prepared by reacting ABTS reagent with manganese dioxide as the oxidising agent. Before analysis, 10 mL of ABTS stock solution was diluted with 25 mL of 0.325 M phosphate buffer and 65 mL of Milli-Q water. An aliquot of each extract (0.05 mL) was mixed with 0.5 mL 0.325 M of phosphate buffer and 1.0 mL of diluted ABTS radical cation solution, and 0.45 mL Milli-Q water was added, to give the final volume of 2 mL. The mixture was shaken and left in the dark for 1 h. The absorbance was measured at 734 nm (UV–Vis spectrophotometer; model 8453; Agilent Technologies, Santa Clara, CA, USA). A lower absorption of the reaction mixture indicates higher free radical scavenging activity. The absorbance of the control was achieved by replacing the seed sample with methanol. The measurement was compared to a standard curve of a Trolox solution, and the radical scavenging activity is expressed as mg TE/g DW. 

### 2.10. Purification of the Seed Extracts

For the purification of the crude seed extracts, 100 mg Strata-X RP cartridges were used (Phenomenex, Torrance, CA, USA). The seed extracts were initially filtered (0.45 µm pore size syringe filters; Chromafil A-45/25; cellulose acetate, hydrophilic membrane; Macherey-Nagel, Düren, Germany), to remove any solid residue. The cartridges were conditioned with 3.0 mL of methanol followed by 3.0 mL of Milli-Q water. Then, 30 mL of the diluted methanol seed extracts (extract:water, 1:9) or 3.0 mL of the hydrolysed seed extracts were applied to the cartridges, and allowed to pass through them. The phenolic compounds remained bound to the cartridges, and the co-extracted compounds were washed from the columns with 4.0 mL of Milli-Q water. The cartridges were then dried using a vacuum pump. The compounds bound to the cartridges were eluted with 2.0 mL of 70% (*v/v*) aqueous methanol. The resulting extracts were filtered through 0.20 µm pore size syringe filters (Chromafil Xtra-20/13; cellulose acetate; Macherey-Nagel, Düren, Germany), and then stored at −80 °C until the liquid chromatography–mass spectrometry (LC-MS) analysis.

### 2.11. Liquid Chromatography–Mass Spectrometry Analysis

Reversed-phase LC-MS analysis was used to separate and quantify the individual phenolic acids in the seed extracts. The LC system used (1100 chromatography system; Agilent Technologies, Santa Clara, CA, USA) included of a thermostated auto-sampler (G1330B), a thermostated column compartment (G1316A), a diode array detector (G1315B), and a binary pump (1312A). The LC system was coupled with a mass spectrometer (Quattro micro API; Waters, Milford, MA, USA). Chromatographic separation was carried out using a C18 column (2.7 μm, 150 mm × 2.1 mm; Ascentis Express) with an C18 guard column (2.7 μm, 5 mm × 2.1 mm; Ascentis Express; Supelco, Bellefonte, PA, USA). The conditions used were: column temperature, 30 °C; injection volume, 20 µL; and mobile phase flow rate, 250 µL/min. The components of the mobile phase were 0.1% aqueous formic acid (solution A) and acetonitrile (solution B). The mobile phase gradient was programmed as follows (%B): 0–2 min, 10%; 2–18 min, 10–60%; 18–18.2 min, 60–80%; 18.2–20 min, 80%; 20–20.2 min, 80–10%; 20.2–26 min, 10%; Detection was performed with the scanning diode array spectra from 240 nm to 650 nm.

The mass spectrometer was operated in negative ionisation mode, and the operating conditions were as follows: electrospray capillary voltage, 3.5 kV; cone voltage, 20 V; extractor voltage, 2 V; source block temperature, 100 °C; desolvation temperature, 350 °C; cone gas flow rate, 30 L/h, and desolvation gas flow rate, 350 L/h. The data signals were acquired and processed on a PC using the MassLynx software (V4.1 2005; Waters Corporation). 

The identification of the individual compounds was achieved by comparing their retention times and both the spectroscopic and mass spectrometric data, with quantification according to peak areas, as compared to previously determined calibration curves. The recoveries of the different compounds were determined using the standard addition method (Appendix A). The samples were spiked with all of the analysed compounds at four spiking levels (1, 10, 200, 2000 µg/g DW of sample) by adding different volumes of a methanolic solution of the analytes.

### 2.12. Statistical Analysis

All of the experiments were performed as six independent replicates, using a complete randomisation method. The data are reported as the means ± standard deviation (SD) for three analyses for each extract. The results were subjected to two-way ANOVA, and the significances of the differences between the mean values were determined using Tukey’s multiple comparison tests. Pearson’s correlation analysis was used to define correlations between the means. All of the tests were performed using the SPSS Statistics software (version 24; IBM, New York, NY, USA). Statistical significance was defined at the level of *p* < 0.05.

## 3. Results and Discussion

### 3.1. Effects of Dehulling and Germination on Growth Rates and Total Dry Weight 

All of the samples of the hulled and cold dehulled seeds reached high germination rates of over 95%. The dry weights of the sprouts were monitored for 96 h, at 24-h intervals, as shown in Figure 1.

As also shown in Figure 1, the mean dry weight losses of the sprouts after 48 h were 1.5% for hulled seeds and 6.4% for dehulled seeds. After 96 h, these losses had reached 8.7% and 21.9%, respectively. Compared to the sprouts from the hulled seeds, the dry weight losses of those from the dehulled seeds were significantly higher at all of the sampled stages of germination (*p* ˂ 0.05). 

To determine the growth of the seedlings, the lengths of the rootlets were measured. After 24 h of germination, the roots were visible for most of the seeds. At this time, the rootlets of hulled seeds were visible as white spots at the tips of the seeds, and were less than 1 mm long, while those of the dehulled seeds were significantly longer, with a mean length of 2.3 mm (*p* < 0.05) (Figure 2) For the rest of the germination period analysed, the rootlets of dehulled seeds were always significantly longer than those of hulled seeds.

This difference in rootlet growth rates between hulled and dehulled seeds is probably due to the higher respiration rate and easier O_2_/CO_2_ transfer for dehulled seeds, compared to hulled seeds. It is also possible that dehulled seeds had better water absorption over the first few hours of germination [42]. A study performed on barley [43] showed that hull-less barley malted more quickly than hulled barley, with significant benefits in terms of reductions in the consumption of water and energy. For these buckwheat seeds, it can be concluded here that the removal of the hulls results in quicker germination, but not significantly greater, germination rates.

It can be assumed that the dry weights of these sprouts will have continued to decrease beyond the 96 h. However, the aim of this study was to determine the dynamic changes in the early germination stages for the production of buckwheat malt that can be used in baking and brewing processes.

### 3.2. Effects of Germination on Total Phenolic Content 

The changes in the content of free and bound phenolic compounds at the different germination stages are shown in Table 1. The total and the free and bound fractions of the phenolic compounds showed dynamic changes. In the first 24 h of germination, no significant increases in the TPC were detected. It is possible that, although the growth and bioactivation of the seeds was observed, some of the water-soluble bioactive compounds might have been lost during the soaking [44]. For the rest of the germination process, gradual and significant increases in TPC were observed, with the maximum values obtained at the end of the 96 h period of monitoring, which is in agreement with a previous study on germinating buckwheat seeds [34]. Compared to hulled seeds, for dehulled seeds the increase in TPC was faster, probably as a consequence of the earlier onset of germination. Germination had the greatest effects on the phenolic content of the free fractions (hulled vs. dehulled, 2.93-fold vs. 5.40-fold increases, respectively). On the other hand, smaller increases were seen for the bound fractions of the phenolic compounds (24% vs. 41% increases, respectively).

For the seeds prior to germination, the contents of free phenolic compounds were similar in hulled and dehulled seeds (Table 1, 0 h). Initially here, as percentages of TPC, the free phenolic compounds were 45.7% in hulled seeds and 54.1% in dehulled seeds. In comparison to cereals (i.e., the *Poaceae* family), the contribution of free phenolic compounds to the TPC in buckwheat seeds is higher, as in cereals most of the phenolic compounds are in a bound form [22].

Throughout the germination period, the bound phenolic content remained significantly higher for hulled seeds compared to dehulled seeds. These data are in agreement with previous studies that reported higher bound phenolic content in buckwheat seed hulls compared to the rest of the seeds [23,45].

From Figure 1, it can be seen that the dry mass of seeds decreases during germination. Here, 10% to 20% of the dry mass was lost during the respiratory metabolic processes required for seed germination and sprout growth to 96 h. The increases in all of the fractions of the phenolic compounds in these sprouts at 96 h will be a combination of de novo synthesis and, to a lesser extent, the concentration of the tissues rich in phenolic compounds, due to the loss of the starchy endosperm [45].

### 3.3. Effect of Germination on Antioxidant Activity

The total AA measured by both the DPPH and ABTS assays gradually increased for both hulled and dehulled sprouts over the 96-h germination period, as summarised in Table 2. 

Before germination, the total AA was higher in hulled seeds compared to dehulled seeds. For both hulled and dehulled seeds, the germination significantly increased AA. The highest measured AA in these methanol extracts was at 96 h of germination for dehulled seeds, and the lowest was for the bound fraction from before germination and after 24 h of germination of dehulled seeds (Table 1). Compared with the Folin–Ciocalteu method for phenolic content (see also Table 1), the AA measures showed similar trends during the course of germination.

In the DPPH test for AA of the free fractions of these seed extracts, up to 48 h after the start of germination there were only minor increases compared to the seeds before germination, with no significant difference between the hulled and dehulled seed extracts. However, after 48 h of germination, AA here increased rapidly, with significantly higher values measured as the maxima that were reached after 96 h. 

These data for the DPPH tests showed significant correlations (*p* ˂ 0.01) between the TPC and AA in the free fractions of both hulled (r = 0.974) and dehulled (r = 0.988) seeds. In the bound fractions, significant correlations were still observed (*p* ˂ 0.05) for both these hulled (r = 0.781) and dehulled (r = 0.926) seed extracts.

The AA obtained using the ABTS assay showed similar trends, with some small differences. Overall, the AA in the free fractions of these seed extracts measured by the ABTS method was significantly higher than that by the DPPH method, which might be explained by the higher activities of the phenolic compounds contributing to the antioxidant potential of the seed extracts towards the ABTS radical [46,47]. It was also noted that the ABTS/DPPH ratio changed, reaching a maximum at 24 h of germination, at around 2.0, and then decreasing to around 1.4 for the 96-h germinated seed extracts. This indicates that during germination, the chemical compositions of the antioxidants were changing, as well as their total concentrations. The different reactivities of particular antioxidants toward the ABTS/DPPH radicals and the changes in their relative compositions will be reflected in these different AA values obtained by the two methods. Indeed, Floegel et al. reported that antioxidant capacities determined for various foods in in vitro assays differed significantly across these two assays (i.e., ABTS and DPPH) [48].

With the ABTS assay, the maximum values of the AA in the free fraction were reached after 96 h of germination, as 19.20 mg TE/g DW and 35.02 mg TE/g DW in hulled and dehulled seeds, respectively. As with the DPPH and Folin-Ciocalteu methods, the lowest values were detected before germination and after 24 h of germination. These data for the ABTS assay also showed significant correlations (*p* ˂ 0.01) between the TPC and AA for the free fractions of both hulled (r = 0.976) and dehulled (r = 0.992) seeds. For the bound fractions, significant correlations were again observed (*p* ˂ 0.05) for the hulled (r = 0.450) and dehulled (r = 0.781) extracts. 

In general, the AA in the bound fractions of these seed extracts were lower compared to the free fractions. In the bound fractions, the differences between the extracts before germination and after 96 h of germination were lower in comparison to the free fractions, where strong increases in the AA were seen during the course of germination.

For the DPPH assay, the lowest values in the bound fractions were seen up to 24 h after the start of germination. Afterwards, the AA began to increase, to reach the maxima at 96 h. In the ABTS assay, the AA of the bound phenolic compounds before germination decreased significantly over the first 24 h of germination (20%, 17% for hulled and dehulled seeds, respectively). Afterwards, the AA slowly increased and reached starting values again after 72 h of germination. These data for the AA in the bound fractions of these seed extracts confirmed the significant differences between these two assays. The lower levels of AA measured using ABTS can be explained by the low content of phenolic compounds that contributed to the antioxidant potential towards the ABTS radical [47]. This is also confirmed by the lower correlations between the AA in the bound fraction in the ABTS assay and the Folin–Ciocalteu and DPPH assays. 

In general, these data for AA also concur with previous studies on buckwheat seeds and germinated buckwheat seeds [24,49,50,51]. However, it needs to be noted that detailed comparisons of the data in the present study with the data in the literature can be difficult, as previous studies were mainly related to buckwheat seeds either before germination [52] or as sprouts [53], without consideration of the effects of dehulling on the levels of the bioactive compounds and the antioxidative potential.

### 3.4. Phenolic Characterisation by Liquid Chromatography–Mass Spectrometry 

Liquid chromatography coupled with mass spectrometry is a powerful tool for the analysis of a wide range of compounds. It is now commonly used for the analysis of phenolics and other bioactive compounds in biological samples. In the present study, LC-MS analysis of the extracts of the germinated seeds (i.e., sprouts) identified several phenolic compounds, most of which were flavonoids in glycosylated and aglycone forms. In contrast to cereals (i.e., the *Poaceae* family) where the majority of phenolic compounds are bound to cell walls [54], the buckwheat seed phenolic compounds are mostly in free forms [52].

The characterisation of the methanol extracts showed that the predominant phenolic compound in the seeds before germination was rutin, with significantly higher levels in hulled seeds compared to dehulled seeds (98.05, 65.39 µg/g, respectively) (Table 2). The other important flavonoids in these seeds were vitexin (33.33, 3.35 µg/g, respectively), catechin (23.15, 10.67 µg/g, respectively) and epicatechin (17.88, 7.48 µg/g, respectively). Three other major phenolic compounds in the germinated seeds, orientin, isoorientin and vitexin, were present at low concentrations before germination, and then greatly increased during germination (Table 2). The higher levels of the phenolic compounds overall in hulled seeds indicate that the majority of these bioactive compounds are stored in the hull and the outer parts of the seed, such that dehulling significantly decreases their levels. Similar data for higher levels of phenolic compounds in buckwheat seed hulls have also been reported in previous studies [23,55].

In the first 24 h of germination, no significant differences were seen for the majority of these phenolic compounds analysed. Interestingly, the rutin levels were even decreased after 24 h, by 13.1% in hulled seeds and 6.7% in dehulled seeds. This decrease can be explained by the leaching of water-soluble phenolic compounds into the steeping water during the early seed soaking [45,56]. Despite the loss of some of these seed bioactive compounds during soaking, with longer germination times there were increased contents of all of these compounds analysed in these sprouts, which is in agreement with the data here for TPC and AA (Table 1).

For the dehulled seeds, in the first 24 h of germination they already showed increased vitexin, catechin and epicatechin levels. These increases would indicate the faster onset of de novo synthesis of these bioactive compounds in the dehulled seeds. As already noted, compared to the hulled seeds, this was probably due to their faster water intake, and consequently more vigorous physiological activity [42,43]. However, the greatest increases in the majority of the compounds analysed here occurred after 48 h of germination. Comparing the levels of phenolics going from 48 h to 72 h of germination, over this 24-h period, rutin levels increased by 1.8-fold and 2.5-fold, vitexin by 6.4-fold and 14-fold, and orientin by 6.9-fold and 5.2-fold in hulled and dehulled seeds, respectively. As can be seen, these relative increases in most of the phenolic compounds were lower in hulled seeds compared to dehulled seeds. During the final 24 h of the germination (i.e., to 96 h), the levels of these bioactive compounds continued to increase, with the maximum values for all of the compounds analysed seen after 96 h of germination. With the exception of hyperin, the levels of all of the compounds analysed were significantly higher in dehulled sprouts (*p* ˂ 0.05).

Overall, the 96-h germination increased the contents of the major phenolics in these seeds as follows: epicatechin, 70-fold; catechin, 44-fold; and rutin, 15-fold. The levels of orientin, isoorientin and vitexin were low in the seeds before germination, and so during germination these levels increased by 878-fold, 1095-fold and 558-fold, respectively. These increases might appear to be particularly large, but it must be noted that these compounds were at particularly low concentrations in the seeds before germination, and their levels were greatly increased by germination. Similar rutin, orientin, isoorientin and vitexin levels in seeds and germinated sprouts have been reported in previous studies [57,58].

Based on the specific reactivities of the phenolic compounds identified [59,60] and their levels in these seed extracts, their contributions to the AA measured can be evaluated. It can be estimated that in the seeds before germination, the contribution of the sum of the phenolic compounds to total AA was relatively low, at only about 5% to 8% of AA determined using DPPH. Other redox-active compounds therefore contributed most to the AA before germination. Upon germination, the levels of the major phenolic compounds increase significantly, which coincided with the increased AA (Figure 3). The calculated relative contributions of the major phenolic compounds after 96 h of germination to the total AA was increased from the few percent beforehand, to approximately 60%. These increases in AA during the germination of these seeds can therefore be attributed preferentially to newly synthesised bioactive phenolic compounds. This is also very important from the nutritional point of view, as the compounds identified here are recognized as having health benefits [16,17,18,19].

Analysis of the bound fractions of these seed extracts showed that p-coumaric acid and vitexin were the major phenolics. Orientin, isoorientin, rutin and hyperin were also present in the bound fractions, although at lower levels than in the free fractions (Table 2). It is of note that the dehulling reduced the bound phenolic content in these seeds, which is in agreement with data from Li et al. [61], where it was shown that the highest levels of bound phenolic compounds are in the seed hulls. The accumulation of bound phenolic compounds during the course of germination was nevertheless similar for hulled and dehulled seeds. As the hull is an inedible part of the seeds, and as it is always removed during processing and preparation for human consumption, this initial loss of phenolic compounds due to the dehulling is not relevant in nutritional terms. 

According to the data in the present study, the dehulling of the seeds prior to germination had significant effects on the levels of these individual phenolic compounds, as well as the phenolic profile in the edible part of these seeds. Most importantly, the content of the major health-promoting compounds, such as orientin, isoorientin, vitexin and rutin, were enhanced more rapidly in dehulled seeds. These data thus show that the combination of cold dehulling and seed germination is a promising method for implementation for the production of new functional food ingredients.

## 4. Conclusions

In the present study, the germination of buckwheat seeds is demonstrated to be an excellent way for increasing their content of phenolic compounds as well as their AA. A new processing approach of the mechanical cold dehulling of buckwheat seeds was used that maintained the high germination rates through seed dehulling. The goal was to determine the effects of this dehulling before germination on seed germination, growth, the TPC, AA and individual phenolic contents. The data obtained show that in comparison with hulled seeds, this cold dehulling maintained high germination rates and promoted faster growth of the buckwheat groats, which resulted in greater increases in TPC, AA and all of the phenolics determined. Some of the important health-promoting compounds of the germinated seeds, including orientin, isoorientin, rutin and vitexin, were detected at greater concentrations in the sprouts from the dehulled seeds. As the inedible part of the seeds was removed (i.e., the hull), such dehulled germinated buckwheat seeds can be used directly in various technological food preparation processes. These improved TPC and antioxidant properties thus further demonstrate that germinated dehulled buckwheat groats provide an excellent raw material for the preparation of functional food products.

## Figures and Tables

**Figure 1 foods-10-00740-f001:**
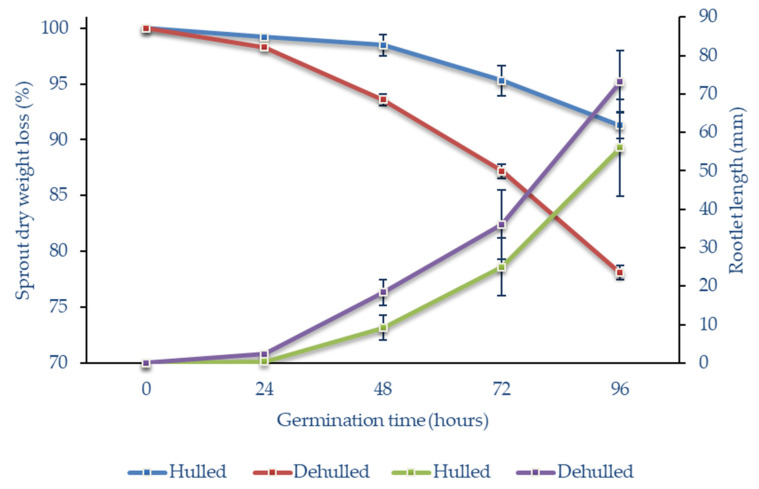
Buckwheat sprout dry weights and rootlet lengths over the 96-h germination period.

**Figure 2 foods-10-00740-f002:**
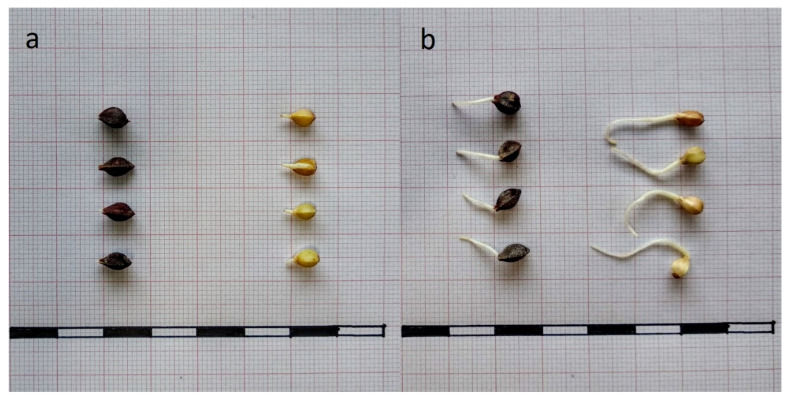
Germinated of representative hulled (left) and dehulled (right) buckwheat seeds after 24 h (**a**) and 48 h (**b**) of germination. Scale measures, 1.0 cm.

**Figure 3 foods-10-00740-f003:**
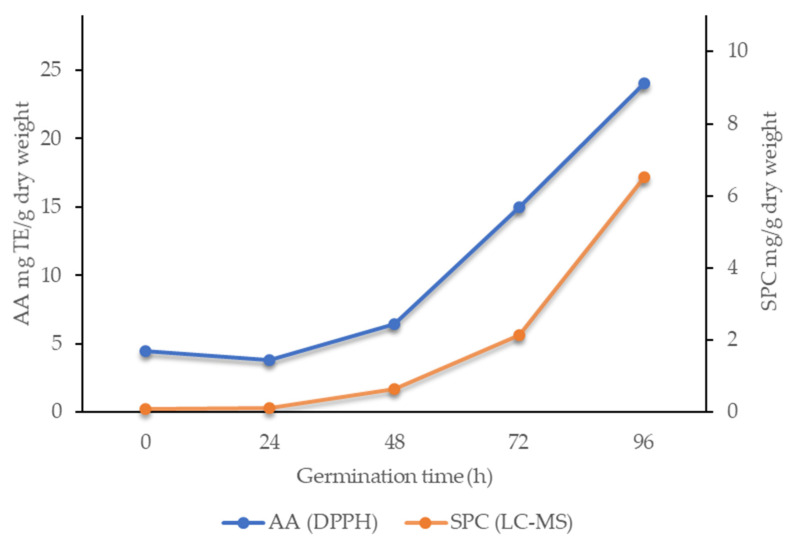
Comparison between antioxidant activity (AA) determined by the DPPH method and the sum of the phenolic compounds (SPC) determined by liquid chromatography-mass spectrometry (LC-MS) for the free fraction of the germinating buckwheat seed extracts.

**Table 1 foods-10-00740-t001:** Total phenolic content (Folin-Ciocalteu) and antioxidant activities (1,1-diphenyl-2-picrylhydrazyl (DPPH), 3-ethylbenzothiazolin-6-sulfonic acid diammonium salt (ABTS)) for the free and bound extraction fractions of the buckwheat seeds during germination.

Analysis	Seed	Measure	Germination Time (h)				
	Preparation		0	24	48	72	96
**Total phenolic content (mg GAE/g dry weight)**							
Folin-	Hulled	Free	4.60 ± 0.42 ^ab^	4.21 ± 0.09 ^a^	5.61 ± 0.38 ^b^	8.01 ± 1.09 ^c^	13.46 ± 0.89 ^d^
Ciocalteu		Bound	5.47 ± 0.29 ^a^	5.87 ± 0.09 ^ab^	6.13 ± 0.32 ^bc^	6.57 ± 0.50 ^c^	7.28 ± 0.37 ^d^
		Total	10.06 ± 0.38 ^a^	10.08 ± 0.19 ^a^	11.74 ± 0.46 ^b^	14.58 ± 1.50 ^c^	20.74 ± 1.20 ^d^
	Dehulled	Free	4.32 ± 0.12 ^a^	4.46 ± 0.15 ^a^	6.36 ± 0.24 ^b^	14.28 ± 1.05 ^c^	23.31 ± 2.09 ^d^
		Bound	3.65 ± 0.12 ^a^	3.91 ± 0.27 ^a^	4.40 ± 0.12 ^b^	4.50 ± 0.25 ^b^	6.19 ± 0.22 ^c^
		Total	7.98 ± 0.21 ^a^	8.37 ± 0.35 ^a^	10.76 ± 0.19 ^b^	18.78 ± 1.03 ^c^	29.50 ± 2.17 ^d^
**Antioxidant activity (TE/g dry weight)**							
DPPH	Hulled	Free	4.61 ± 0.19 ^a^	3.99 ± 0.08 ^a^	5.45 ± 0.56 ^a^	7.60 ± 0.75 ^b^	14.00 ± 2.14 ^c^
		Bound	4.15 ± 0.38 ^a^	4.04 ± 0.42 ^a^	5.64 ± 0.66 ^b^	6.10 ± 1.19 ^b^	8.53 ± 0.53 ^c^
		Total	8.76 ± 0.49 ^a^	8.03 ± 0.38 ^a^	11.08 ± 1.03 ^b^	13.70 ± 1.83 ^c^	22.53 ± 2.49 ^d^
	Dehulled	Free	4.45 ± 0.15 ^ab^	3.84 ± 0.18 ^a^	6.42 ± 0.71 ^b^	15.00 ± 1.06 ^c^	24.07 ± 2.39 ^d^
		Bound	2.01 ± 0.22 ^a^	2.54 ± 0.19 ^a^	3.37 ± 0.27 ^b^	3.99 ± 0.66 ^c^	5.80 ± 0.16 ^d^
		Total	6.46 ± 0.33 ^a^	6.38 ± 0.31 ^a^	9.79 ± 0.64 ^b^	18.99 ± 1.41 ^c^	29.87 ± 2.29 ^d^
ABTS	Hulled	Free	7.02 ± 0.18 ^a^	7.53 ± 0.36 ^a^	9.09 ± 0.35 ^b^	12.34 ± 0.82 ^c^	19.20 ± 1.11 ^d^
		Bound	5.94 ± 0.65 ^b^	4.72 ± 0.47 ^a^	4.64 ± 0.54 ^a^	5.06 ± 0.46 ^a^	6.65 ± 0.33 ^b^
		Total	12.96 ± 0.61 ^a^	12.26 ± 0.48 ^a^	13.74 ± 0.71 ^a^	17.4 ± 1.17 ^b^	25.85 ± 1.31 ^c^
	Dehulled	Free	7.19 ± 0.22 ^a^	7.73 ± 0.38 ^ab^	10.44 ± 1.22 ^b^	23.22 ± 2.28 ^c^	35.02 ± 2.74 ^d^
		Bound	3.14 ± 0.26 ^a^	2.59 ± 0.25 ^a^	2.45 ± 0.23 ^a^	2.80 ± 0.26 ^a^	4.71 ± 1.02 ^b^
		Total	10.34 ± 0.46 ^a^	10.32 ± 0.6 ^a^	12.89 ± 1.28 ^a^	26.01 ± 2.36 ^b^	39.73 ± 3.51 ^c^

Data are means ± SD, from three independent replicates. Means with different letters indicate statistically significant differences between the different stages of germination (*p* ˂ 0.05). GAE: gallic acid equivalents; TE: Trolox equivalents.

**Table 2 foods-10-00740-t002:** Contents of the individual phenolic compounds (µg/g dry weight (DW)) in the free and bound fractions of the germinating buckwheat seeds.

Phenolic	Phenolic Content (µg/g DW) during Germination (h) for the Hulled and Dehulled Seeds									
Compound	0		24		48		72		96	
	Hulled	Dehulled	Hulled	Dehulled	Hulled	Dehulled	Hulled	Dehulled	Hulled	Dehulled
**Free fraction**										
Orientin	11.82 ± 0.45 ^b^	2.51 ± 0.12 ^a^	13.54 ± 1.63 ^b^	2.54 ± 0.31 ^a^	24.96 ± 3.97 ^a^	96.78 ± 7.57 ^b^	171.98 ± 17.67 ^a^	501.34 ± 33.12 ^b^	865.85 ± 91.02 ^a^	2205.06 ± 213.18 ^b^
Isoorientin	7.79 ± 0.37 ^b^	1.28 ± 0.07 ^a^	9.43 ± 0.98 ^b^	1.25 ± 0.17 ^a^	11.88 ± 1.02 ^a^	35.33 ± 4.46 ^b^	76.26 ± 4.97 ^a^	339.68 ± 34.49 ^b^	725.37 ± 35.5 ^a^	1402.5 ± 106.51 ^b^
Rutin	98.05 ± 8.66 ^b^	65.39 ± 5.67 ^a^	85.17 ± 5.42 ^b^	60.96 ± 5.63 ^a^	106.04 ± 8.11 ^a^	138.46 ± 7.76 ^b^	194.76 ± 13.56 ^a^	347.6 ± 51.67 ^b^	664.32 ± 67.18 ^a^	967.66 ± 81.34 ^b^
Vitexin	33.33 ± 3.3 ^b^	3.35 ± 0.33 ^a^	52.13 ± 2.73 ^b^	7.99 ± 0.68 ^a^	59.91 ± 4.55 ^a^	51.55 ± 3.72 ^b^	385.83 ± 19.31 ^a^	723.34 ± 60.09 ^b^	1180.89 ± 157.33 ^a^	1869.17 ± 185.14 ^b^
Catechin	23.15 ± 1.32 ^b^	10.67 ± 0.95 ^a^	18.55 ± 1.48 ^a^	27.39 ± 2.5 ^b^	64.5 ± 8.26 ^a^	148.41 ± 10.65 ^b^	134.95 ± 6.68 ^a^	285.1 ± 17.96 ^b^	243.85 ± 21.19 ^a^	472.5 ± 33.23 ^b^
Epicatechin	17.88 ± 1.15 ^b^	7.48 ± 1.22 ^a^	12.05 ± 1.23 ^a^	32.84 ± 1.94 ^b^	46.89 ± 3.76 ^a^	148.59 ± 6.72 ^b^	89.44 ± 2.70 ^a^	413.11 ± 15.19 ^b^	150.16 ± 20.53 ^a^	527.6 ± 39.57 ^b^
Hyperin	13.48 ± 1.33 ^b^	1.72 ± 0.09 ^a^	8.15 ± 1.66 ^b^	0.87 ± 0.06 ^a^	10.00 ± 0.94 ^b^	1.78 ± 0.27 ^a^	9.90 ± 0.83 ^a^	8.23 ± 1.47 ^b^	20.14 ± 2.68 ^a^	19.66 ± 1.41 ^a^
p-Coumaric acid	ND	ND	1.64 ± 0.11 ^a^	2.93 ± 0.27 ^b^	5.33 ± 0.35 ^a^	13.26 ± 1.22 ^b^	11.97 ± 1.29 ^a^	16.8 ± 2.05 ^b^	18.62 ± 1.62 ^a^	31.38 ± 2.06 ^b^
**Bound fraction**										
Orientin	2.12 ± 0.18	ND	1.09 ± 0.13	ND	1.17 ± 0.09 ^b^	0.05 ± 0.01 ^a^	1.48 ± 0.13 ^b^	0.17 ± 0.01 ^a^	1.84 ± 0.14 ^a^	3.89 ± 0.22 ^b^
Isoorientin	0.85 ± 0.07 ^b^	0.21 ± 0.03 ^a^	0.94 ± 0.08 ^b^	0.39 ± 0.03 ^a^	1.06 ± 0.09 ^b^	0.49 ± 0.03 ^a^	1.25 ± 0.09 ^a^	1.52 ± 0.11 ^b^	3.07 ± 0.41 ^a^	11.63 ± 1.13 ^b^
Rutin	0.34 ± 0.03 ^b^	0.19 ± 0.01 ^a^	0.12 ± 0.01	ND	0.34 ± 0.04 ^a^	0.51 ± 0.06 ^b^	0.49 ± 0.04 ^a^	0.74 ± 0.09 ^b^	0.61 ± 0.05 ^a^	0.86 ± 0.11 ^b^
Vitexin	17.99 ± 1.24 ^b^	0.47 ± 0.03 ^a^	13.74 ± 0.97 ^b^	0.53 ± 0.04 ^a^	22.86 ± 2.55 ^b^	0.93 ± 0.07 ^a^	25.41 ± 2.31 ^b^	5.31 ± 0.49 ^a^	35.95 ± 2.33 ^b^	18.47 ± 1.14 ^a^
Catechin	ND	ND	ND	ND	ND	ND	ND	ND	ND	ND
Epicatechin	ND	ND	ND	ND	ND	ND	ND	ND	ND	ND
Hyperin	1.25 ± 0.03	ND	0.98 ± 0.07	ND	1.66 ± 0.15	ND	3.07 ± 0.23	ND	4.24 ± 0.57	ND
p-Coumaric acid	31.32 ± 0.25 ^b^	2.64 ± 0.14 ^a^	32.79 ± 2.15 ^b^	3.39 ± 0.32 ^a^	36.82 ± 2.87 ^b^	4.19 ± 0.37 ^a^	37.91 ± 2.86 ^b^	5.06 ± 0.41 ^a^	60.36 ± 7.24 ^b^	7.36 ± 0.63 ^a^

Data are means ± SD, from three independent replicates. Means with different letters in rows indicate statistically significant differences between contents in the same stage of germination (hulled vs. dehulled) (*p* ˂ 0.05); not detected (ND).

## Data Availability

Not applicable.

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
