# Peer review of "Germinated Buckwheat: Effects of Dehulling on Phenolics Profile and Antioxidant Activity of Buckwheat Seeds"

_foods, 2021, doi:10.3390/foods10040740_

Round 1
Reviewer 1 Report
Abstract. Please use either only buckwheat or buckwheat seeds. The use of both terms is very confusing.
Line 15. Please avoid the use of “we” in the abstract and in the manuscript. It is not usual and in my opinion appropriate.
Abstract. Extensive revision is needed. Two time the term LC-MS is reported with no meaning. It is not important to write in the abstract that the methods ABTS, DPPH, folin etc.
Lines 32-34. In recent years… but the reference is not recent. Please consider https://doi.org/10.3390/foods9060790
I advise the authors to consider the addition and discussion of the following relevant works (especially replacing some references in the introduction):
https://doi.org/10.3390/foods5020027
https://doi.org/10.3390/foods9030296
https://doi.org/10.3390/foods9060790
https://doi.org/10.3390/foods9081130
https://doi.org/10.3390/foods9091218
Lines 261-263. Delete
Line 272. Do not start a sentence with numbers. E.g. 24
Table 1. Lines 298-299. I cannot understand why to compare statistically the differences between a faction and total phenolic content. It is not appropriate, but also negatively affects the statistical analysis.
Line 307. Delete reference.
Section 3.3. It should be reduced. Extensive repetition of the results. The results are also presented in table 2. Lines 388-399. The authors are just reporting the values of other studies without meaning since as they also report is difficult to compare.
Too many figures tables please reduce.
The novelty of the present study is limited or at least not clarified by the authors. What about the following works?
https://doi.org/10.1111/1750-3841.12830
https://doi.org/10.1007/s13197-011-0316-1
https://doi.org/10.1016/j.jff.2014.01.031
In general the work has limited novelty and according to authors the novelty is limited to the use of local variety (lines 88-89).
Author Response
Reviewer #1
Comments and Suggestions for Authors:
- Please use either only buckwheat or buckwheat seeds. The use of both terms is very confusing.
Authors’ reply: We thank the Reviewer for this comment. In the full revision and English editing now carried out on the revised manuscript, these terms are now used in their strictest senses. Therefore, “buckwheat” alone now refers only to the plant, various forms of “buckwheat seeds”, “buckwheat sprouts” and others are specifically used where other plants are also indicated (i.e., where specification of “buckwheat” is needed for full understanding), and in all other places just “seeds” are referred to, along with terms like “germinated seeds”, “sprouts” without the use of “buckwheat” all the time. We believe (as our language editor tells us) that this will have removed the problem here.
- Line 15. Please avoid the use of “we” in the abstract and in the manuscript. It is not usual and in my opinion appropriate.
Authors’ reply: As suggested by the Reviewer, all sentences containing the term “we” have been changed to more appropriate forms.
- Extensive revision is needed. Two time the term LC-MS is reported with no meaning. It is not important to write in the abstract that the methods ABTS, DPPH, folin etc.
Authors reply: The Abstract has been improved. We have removed some method names and also added additional information about the most important results.
- Lines 32-34. In recent years… but the reference is not recent. Please consider
https://doi.org/10.3390/foods9060790
I advise the authors to consider the addition and discussion of the following relevant works (especially replacing some references in the introduction):
https://doi.org/10.3390/foods5020027
https://doi.org/10.3390/foods9030296
https://doi.org/10.3390/foods9060790
https://doi.org/10.3390/foods9081130
https://doi.org/10.3390/foods9091218
Authors’ reply:
We thank the Reviewer for these suggestions, and additional references have now been added as relevant. Along with suggested references, some of the older literature has also been supplemented with new scientific reports (Ref. no. 14, 28, 32).
- Lines 261-263. Delete
Authors’ reply: As suggested by the Reviewer, the mentioned lines have been deleted from the manuscript.
- Line 272. Do not start a sentence with numbers. E.g. 24
Authors’ reply: As suggested by the Reviewer, the proposed change has been applied.
»After 24 h of germination, the roots were visible for most of the seeds.«
We note also a couple of other instances of the same in the manuscript that have now been picked up and corrected by our language editor.
- Table 1. Lines 298-299. I cannot understand why to compare statistically the differences between a faction and total phenolic content. It is not appropriate, but also negatively affects the statistical analysis.
Authors’ reply: We thank the Reviewer for this suggestion, and we agree that the vertical comparison between the fractions was not relevant, so it has been removed from the Table.
- Line 307. Delete reference.
Authors’ reply: Indeed, the reference mentioned was inappropriately inserted. We have changed the line so that the reference is now properly cited.
»For the rest of the germination process, gradual and significant increases in TPC were observed, with the maximum values obtained at the end of the 96 h period of monitoring, which is in agreement with a previous study on germinating buckwheat seeds [34].«
- Section 3.3. It should be reduced. Extensive repetition of the results. The results are also presented in table 2. Lines 388-399. The authors are just reporting the values of other studies without meaning since as they also report is difficult to compare.
Authors’ reply: As suggested by the Reviewer, we have removed some of these results and most of the comparison with results from previous studies. We agree with the Reviewer that direct comparisons of these data may be difficult.
Deleted lines: Lines (324-326), (340-341), (373-374), (382-392)
- Too many figures tables please reduce.
Authors’ reply: Changes in the Tables has been applied. Table 1/ Table 2 and Table 3/ Table 4 have been merged so now the final number of Tables is two. We also note that our language editor has provided suggestions for the better structuring of the Tables, which we feel the Reviewers will also find to be of benefit.
- The novelty of the present study is limited or at least not clarified by the authors. What about the following works?
https://doi.org/10.1111/1750-3841.12830
https://doi.org/10.1007/s13197-011-0316-1
https://doi.org/10.1016/j.jff.2014.01.031
- In general the work has limited novelty and according to authors the novelty is limited to the use of local variety (lines 88-89).
Authors’ reply to comments 11 and 12:
The novelty of our study coincides with the topics of the special issue: Functional Cereal Foods for Health Benefits: Genetic and/or Processing Strategies to Enhance the Quali-Quantitative Composition of Bioactive Components. Cereal dehulling is a mechanical or hydrothermal process in which the grains are often damaged and therefore the germination rate is low. In our study, a new processing approach of mechanical – the cold dehulling method – was used for the buckwheat which maintained a high germination rate. When this dehulled buckwheat was germinated it provided large increases in dietary phenolic compounds. Previous studies (and those mentioned here) and other scientific articles have focused on the effects of germination on phenolic content and antioxidant activity of whole germinated buckwheat. To the best of our knowledge, there are no data available for phenolics in dehulled germinated buckwheat. Since the inedible part of the grain is already removed, dehulled germinated buckwheat can be used directly in various food preparation technological processes. The results of our experiment also show the phenolic composition of the ‘Čebelica’ cultivar, for which no data have been published to date.
Reviewer 2 Report
The paper presents effect of dehulling on phenolic profile and antioxidant activity of buckwheat seeds. Manuscript concerns an interesting issue that may be the basis for the development of valuable scientific material. After a careful survey, I came to the conclusion that the manuscript is well written.
1. The abstract does not present the most important results and conclusions from the conducted research. It needs to be changed.
2. What were purity of standards used in experiment?
3. How did the Authors inspect, that the extraction was exhaustive?
4. There is no information about validation of chromatographic methods.
5. How did Authors inspect the purity of peaks?
6. Exemplary chromatograms of analyzed compounds should be provided.
7. Many items in the literature are out of date. Please, supplement the literature with new scientific reports.
8. The conclusions do not relate to the obtained research results. They do not specify how the research hypotheses were verified. They needs to be rewritten.
Author Response
Reviewer #2
Comments and Suggestions for Authors
The paper presents effect of dehulling on phenolic profile and antioxidant activity of buckwheat seeds. Manuscript concerns an interesting issue that may be the basis for the development of valuable scientific material. After a careful survey, I came to the conclusion that the manuscript is well written.
Authors’ reply: We thank the Reviewer for the recognition of the value of our study, and for the time and effort that was obviously put in to help us to improve manuscript. We have done our best to integrate following comments into the overall concept of our manuscript revision.
- The abstract does not present the most important results and conclusions from the conducted research. It needs to be changed.
Authors’ reply: As suggested by the Reviewer, the Abstract has now been improved. We have removed some method names and also added additional information about the most important results.
- What were purity of standards used in experiment?
Authors’ reply: In section 2.1 we have added the product numbers (PN) of the standard compounds used from which all of the information can be retrieved. Also, we have added following line:
“All of the standards used were analytical or HPLC grade.” Line 111
- How did the Authors inspect, that the extraction was exhaustive?
Authors’ reply: In section 2.11. LC-MS analysis, a paragraph has been added with a better description of how the recoveries for the whole method were determined (as cited in Authors’ reply to comment number 4.). In this field of determination of phenolic compounds, we have previous experience where the methods have been thoroughly investigated:
10.1016/j.foodchem.2016.04.030
10.1016/j.indcrop.2020.112851
10.1016/j.foodchem.2011.09.033
10.3390/molecules22030375
10.1002/cbdv.201100337
10.1016/j.foodchem.2011.02.077
- There is no information about validation of chromatographic methods.
Authors’ reply: In the Methods section, we have added a paragraph (Lines 240-244) with the method recoveries, and an additional Table with the recoveries for the individual compounds (Supplementary Materials Table S1).
“The recoveries of the different compounds were determined using the standard addition method (Supplementary Materials Table S1). The samples were spiked with all of the analysed compounds at four spiking levels (1, 10, 200, 2000 µg/g DW of sample) by adding different volumes of a methanolic solution of the analytes.“
- How did Authors inspect the purity of peaks?
Authors’ reply: Peak purity was investigated by comparisons of spectra recorded with a diode-array detector and mass spectrometric data (MS and MS/MS fragmentation).
- Exemplary chromatograms of analysed compounds should be provided.
Authors’ reply: Representative chromatograms are now included in the Supplementary Materials (Figures S1).
- Many items in the literature are out of date. Please, supplement the literature with new scientific reports.
Authors’ reply: As suggested by the Reviewer, additional references have been added. Most of the older literature has been supplemented with new ones.
References added/replaced: No: 2, 3, 4, 6, 7, 8, 9, 14, 28, 32.
Reference No. 42: Although this is old, it is part of the rare literature available to us that covers the field of water-sensitivity and oxygen availability of germinated seeds.
- The conclusions do not relate to the obtained research results. They do not specify how the research hypotheses were verified. They need to be rewritten.
Authors’ reply: As suggested by the Reviewer, the Conclusions have been rewritten: additional information added.
“In the present study, germination of buckwheat seeds is demonstrated to be an excellent way for increasing their content of phenolic compounds as well as their AA. A new processing approach of mechanical cold dehulling of the buckwheat seeds was used that maintained the high germination rates through seed dehulling. The goal was to determine the effects of this dehulling before germination on seed germination, growth, TPC, AA and individual phenolic contents. The data obtained show that in comparison with hulled seeds, this cold dehulling maintained high germination rates and promoted faster growth of the buckwheat groats, which resulted in greater increases in TPC, AA and all of the phenolics determined. Some of the important health-promoting compounds of the germinated seeds, including orientin, isoorientin, rutin and vitexin, were detected at greater concentrations in the sprouts from the dehulled seeds. As the inedible part of the seeds was removed (i.e., the hull), such dehulled germinated buckwheat seeds can be used directly in various technological food preparation processes. These improved TPC and antioxidant properties thus further demonstrate that germinated dehulled buckwheat groats provide an excellent raw material for preparation of functional food products.”
Round 2
Reviewer 1 Report
The manuscript has been improved following the reviewers' comments.
My comments have been replied.
Regarding the novelty of the manuscript I have some concerns. However, I advise the authors to add some text from their reply in my novelty comment in the introduction just before the material and methods section (for example in lines 84-88). This will provide the reader the aim of the study and its novelty.
Author Response
Reviewer #1
Comments and Suggestions for Authors:
- (x) English language and style are fine/minor spell check required
Authors’ reply: Additional spell check has been performed and minor corrections have been applied.
- Regarding the novelty of the manuscript I have some concerns. However, I advise the authors to add some text from their reply in my novelty comment in the introduction just before the material and methods section (for example in lines 84-88). This will provide the reader the aim of the study and its novelty.
Authors’ reply: We thank the Reviewer for this comment. Additional text (lines 85-88) from our reply has been added in manuscript for better recognition of the aim and novelty of our study.